# Antifungal Effect of Magnolol and Honokiol from *Magnolia officinalis* on *Alternaria alternata* Causing Tobacco Brown Spot

**DOI:** 10.3390/molecules24112140

**Published:** 2019-06-06

**Authors:** Ya-Han Chen, Mei-Huan Lu, Dong-Sheng Guo, Ying-Yan Zhai, Dan Miao, Jian-Ying Yue, Chen-Hong Yuan, Ming-Min Zhao, De-Rong An

**Affiliations:** 1College of Plant Protection and State Key Laboratory of Crop Stress Biology for Arid Areas, Northwest A&F University, Yangling 712100, China; yhchen1018@nwafu.edu.cn (Y.-H.C.); lu_meihuan@sina.com (M.-H.L.); gds1995908@163.com (D.-S.G.); zyyzhaiyingyan@163.com (Y.-Y.Z.); ych951008@163.com (C.-H.Y.); 2College of agriculture, Inner Mongolia Agricultural University, Huhhot 010018, China; miaodan0115@163.com (D.M.); yuejianying2018@163.com (J.-Y.Y.); 3Microbial Resources of Research Center, Microbiology Institute of Shaanxi, Xian 710043, China

**Keywords:** *Magnolia officinalis*, magnolol, honokiol, antifungal activity, *Alternaria alternata*

## Abstract

In this study, two phenol compounds, magnolol and honokiol, were extracted from *Magnolia officinalis* and identified by LC-MS, ^1^H- and ^13^C-NMR. The magnolol and honokiol were shown to be effective against seven pathogenic fungi, including *Alternaria alternata* (Fr.) Keissl, *Penicillium expansum* (Link) Thom, *Alternaria dauci* f.sp. solani, *Fusarium moniliforme* J. Sheld, *Fusarium oxysporum* Schltdl., *Valsa mali* Miyabe & G. Yamada, and *Rhizoctonia solani* J.G. Kühn, with growth inhibition of more than 57%. We also investigated the mechanisms underlying the potential antifungal activity of magnolol and honokiol. The results showed that they inhibited the growth of *A. alternata* in a dose-dependent manner. Moreover, magnolol and honokiol treatment resulted in distorted mycelia and increased the cell membrane permeability of *A. alternata*, as determined by conductivity measurements. These results suggest that magnolol and honokiol are potential antifungal agents for application against plant fungal diseases.

## 1. Introduction

*Magnolia officinalis* is an important herb in traditional Chinese and Japanese medicine for treating conditions such as gastrointestinal disorders, thrombotic stroke, allergic disease, typhoid fever, anxiety, and nervous disturbance [1]. *M. officinalis* contains a diverse group of biologically active compounds, including magnolol, honokiol, 4-O-methylhonokiol, obovatol, and other neolignan compounds [2,3]. Magnolol and honokiol are the two major secondary metabolites found in the ethanolic extracts of Chinese *M. officinalis* at concentrations of 1.25% and 1.81%, respectively [4,5]. Several reports showed that magnolol and honokiol possess antibacterial, anti-inflammatory, antifungal, anti-cancer, proapoptotic, and anti-osteoclastogenetic activities [3,6,7,8,9,10,11]. For example, they exhibit potent antifungal activity against human fungal pathogens, such as *Trichophyton mentagrophytes*, *Candida albicans*, and *Cryptococcus neoformans* [4,5,6,12,13,14].

Tobacco is an important economic plant worldwide for cigarette production. *Alternaria alternata* (Fr.) Keissl is the causal agent of tobacco brown spot, which is one of the most harmful diseases of tobacco, causing huge crop losses around the world [15,16,17]. In China, the annual incidence area of *A. alternata* is about 100,000 hm^2^, leading to economic losses [8]. Infection of *A. alternata* leads to the appearance of small yellowish-brown round spots on the tobacco leaves. Initially, the spots are blade-shaped and then change into a larger circle [18], which could significantly affect the quantity and quality of tobacco [19,20,21]. To control this disease, chemical reagents, for example, dimetachlone, have been widely used in the field [20]. However, chemical agents have the potential to develop fungicide resistance in plant pathogens and cause environmental pollution, resulting in serious health risks to animals and human beings. Indeed, many countries have restricted the application of chemical fungicides to control plant diseases [22]. Therefore, the plant-derived fungicides are promising alternatives, since they are mostly non-phytotoxic, systematic, readily biodegradable, and environmentally safe.

In recent years, many studies have reported the antifungal activity of plant-derived fungicides against *A. alternata*. Plumbagin, a type of naphthoquinone, isolated from leaves of *Nepenthes ventricosa*, inhibited the growth of *A. alternata* [23]. Jing et al. have found that eugenol, an essential oil from the buds of *Syringa oblata* Lindl, exhibited the complete inhibition of mycelial growth of *A. alternata*, suggesting its potential application as a fungicide [21].

The extracts of *M. officinalis* showed potent antifungal activity against many plant pathogenic fungi. However, its antifungal activity against *A. alternata* has not been reported. Here, we identify two compounds, magnolol and honokiol, from *M. officinalis*. We found that these two active compounds have the potential to inhibit mycelium growth effectively and to increase cell membrane permeability of *A. alternata*. Their antifungal effects on *Penicillium expansum* (Link) Thom, *Alternaria dauci* f.sp. solani, *Fusarium moniliforme* J. Sheld, *Fusarium oxysporum* Schltdl., *Valsa mali* Miyabe & G. Yamada, and *Rhizoctonia solani* J.G. Kühn were also investigated in vitro.

## 2. Results and Discussion

### 2.1. Identification of Magnolol and Honokiol from *M. officinalis*

The purities of compounds magnolol and honokiol were identified using high-performance liquid chromatography (HPLC). As shown in Figure 1, their purities were higher than 98%. The structures of magnolol and honokiol were then confirmed by ^1^H-NMR, ^13^C-NMR (Figure 2), and HP-MS spectra (Figure 3), respectively. Their spectral data were identical to that of the previously reported literature [24,25,26].

Magnolol (5,5′-diallyl-2,2′-dihydroxybiphenyl): colorless solid; ^1^H-NMR (500 MHz, CD_3_OD): δ (ppm) 3.35 (d, *J* = 6.7 Hz, 4H), 4.86 (s, 2H), 5.02–5.10 (m, 4H), 5.95–6.03 (m, 2H), 6.88 (d, *J* = 8.2 Hz, 2H), 7.04–7.06 (m, 4H); ^13^C-NMR (125 MHz, CD_3_OD): δ (ppm) 40.3, 115.5, 117.4, 127.6, 129.7, 132.6, 133.1, 139.3, 153.2; HR-MS (ESI): *m*/*z* calcd for C_18_H_18_O_2_ ([M + H]^+^) 267.1378, found 267.1382.

Honokiol (5,5′-diallyl-2,4′-dihydroxybiphenyl): colorless solid; ^1^H-NMR (500 MHz, CD_3_OD): δ (ppm) 3.30 (d, *J* = 6.7 Hz, 2H), 3.41 (d, *J* = 6.5 Hz, 2H), 4.86 (s, 2H), 5.00–5.11 (m, 4H), 5.93–6.09 (m, 2H), 6.79–7.26 (m, 6H); ^13^C-NMR (125 MHz, CD_3_OD): δ (ppm) 35.3, 40.4, 115.3, 115.5, 116.8, 127.3, 128.7, 129.1, 129.9, 131.4, 131.5, 131.9, 132.5, 138.4, 139.5, 153.2, 155.5; HR-MS (ESI): *m*/*z* calcd for C_18_H_18_O_2_ ([M + H]^+^) 267.1378, found 267.1373.

Given the fact that usage of synthetic and mechanical fungicides have associated adverse effects, it is important to develop environmentally friendly and biodegradable botanical fungicides to produce green and safe food products for animal and human consumption [22].

### 2.2. Antifungal Activity of Magnolol and Honokiol against A. alternata

Magnolol and honokiol inhibited the mycelial growth of *A. alternata* in a dose-dependent manner (Table 1). The mycelial growth was noticeably reduced at higher concentrations of magnolol and honokiol, indicating that magnolol and honokiol had an inhibitory effect on *A. alternata*. The growth inhibition of *A. alternata* was 77% and 91% in the presence of 0.1 mg/mL of magnolol and honokiol, respectively. The antifungal activity of carbendazim was much higher than that of magnolol and honokiol (Table 1), when used at concentration with 0.001 mg/mL. At concentrations greater than or equal to 3 mg/mL, magnolol completely inhibited the growth of *A. alternata*, whereas greater than or equal to 1 mg/mL of honokiol completely inhibited growth. In addition, we found that there was a significant positive correlation between the concentration of magnolol and honokiol, and the inhibitory effects on mycelial growth of *A. alternata*.

To the best of our knowledge, the antifungal activity of magnolol and honokiol against *A. alternata* in vitro has not yet been reported. However, several studies have been conducted to evaluate their antifungal and antibacterial activities. For instance, magnolol and honokiol could effectively inhibit the growth of *Pyricularia oryzae*, *Phytophthora infestans*, *Puccinia recondita*, and *Colletotrichum capsici* in plants [15].

Carbendazim is a systemic fungicide with both protective and curative activities against a wide range of fungal diseases [27]. Its antifungal activity was much higher than that of magnolol and honokiol (Table 1), when used at concentrations of 0.001 mg/mL. However, it has been reported that carbendazim, an endocrine disrupter, has mutagenic and teratogenic effects on mammals at low doses [28].

The mycelial radial growth inhibition rates were 77% and 91% for magnolol and honokiol, respectively, when the concentration of each was 0.1 mg/mL. Moreover, the antifungal activity was much higher than eugenol (0.1 mg/mL), which showed 11% inhibition of mycelial growth. On the basis of the present results, which are also in agreement with the reported by Edris and Farrag (2003), eugenol was completely inactive on *Sclerotinia sclerotiorum* (Lib), *Rhizopus stolonifer* (Ehrenb. exFr) *Vuill* and *Mucor* sp. (Fisher), but mixing eugenol and linalool in a ratio similar to their concentrations in the original oil was found to enhance the antifungal properties oil indicating a synergistic effect [29].

Concentrations greater than or equal to 3 mg/mL of magnolol completely inhibited the growth of *A. alternata*, whereas greater than or equal to 1 mg/mL of honokiol completely inhibited the growth. However, no toxicity tests were done on other eukaryotic cells. Concentrations higher than 0.1 mg/mL are too high to show selective toxicity. We evaluated the growth inhibition up to 0.1 mg/mL of both magnolol and honokiol, which caused 77% inhibition in fungal colonies, and this inhibition is sufficient and close to the MIC80 (the minimum inhibitory concentration 80% of microbial growth) value of the antifungal activity of natural products with good results.

### 2.3. Synergistic Effect of Magnolol and Honokiol

In order to examine the synergistic effect of a combination of two compounds on the mycelial growth of *A. alternata*, we combined two compounds at different ratios and tested their effect with in vitro assays (Table 2). The results showed that the mycelial growth inhibition was higher than the other six volume ratios, with a radial growth inhibition of 91%, under the condition of the volume ratio of magnolol and honokiol (0:1). When the volume ratio of magnolol and honokiol was 1:4, a growth inhibition of 89% of mycelial growth was inhibited. When the volume ratio was 1:0, the growth inhibition was only 77%. The results showed that there was not any synergic effect of these two compounds.

### 2.4. Growth Inhibition and the Effect on Morphology

At 7 days post-inoculation, we measured the diameter of the colonies of *A. alternata* on PDA (potato dextrose agar) plates containing 0.1 mg/mL magnolol or honokiol (Figure 4b,c). We observed an apparent decrease in colony size in the PDA plate containing either magnolol (Figure 4b) or honokiol (Figure 4c) when compared with the control PDA plate (Figure 4a). Subsequently, the effect of magnolol and honokiol on the inhibition of hyphae development was also evaluated by examining the morphology of *A. alternata* using scanning electron microphotography (SEM). Figure 4d–f show the micrographs of mycelia from SEM analysis. Our results indicate that, in the control plate, fungal hyphae appeared normal and cylindrical with a smooth surface (Figure 4d). However, treatment of magnolol and honokiol led to morphological changes in the hyphae of *A. alternata*. In particular, the mycelia were curly, distorted, aggregated, and partly squashed, when grown in the presence of magnolol (Figure 4e). In the case of honokiol treatment, shrunken and distorted mycelia of *A. alternata* were observed (Figure 4f).

These results are consistent with the previous study that many changes occur in the morphology of fungal hyphae after treatment with chemical fungicides, chitosan, and natural botanicals [27,28,29]. We assume that the mycelium changes may generally occur because of increased permeability of cells, which resulted in the leakage of small molecular substances, ions, lesions, and disturbances in cell metabolism [30]. The marked leakage of cytoplasmic material is commonly used as an indication of gross and irreversible damage of the cytoplasmic and plasma membranes [30,31,32,33].

### 2.5. Effects of Magnolol and Honokiol on Cell Membrane Permeability of A. alternata

To address whether both magnolol and honokiol could inhibit the hyphae growth by affecting the cell membrane permeability of *A. alternata*, we tested the effect of magnolol (0.1 mg/mL) and honokiol (0.1 mg/mL) on the cell membrane permeability of *A. alternata* for a period of 0–5 h (Figure 5). We showed that the cell membrane permeability was significantly increased (*P* < 0.05) when *A. alternata* was grown on a PDA plate containing magnolol and honokiol at different time points. The conductivity of the *A. alternata* suspensions after treatment with magnolol for 1 h was 167.34 *S*/cm, which is much higher than that of the control, 102.77 *S*/cm. The conductivity of the fungal suspension after treatment with honokiol was 192.81 *S*/cm.

These findings indicate that the antifungal activity of magnolol and honokiol may be associated with the irreversible damage to the cytoplasmic membranes of *A. alternata*, which results in ion leakage from the cells and an imbalance in osmotic pressure of the intra- and extra-cellular membranes.

### 2.6. The Inhibitory Efficiency of Magnolol and Honokiol on Six Phytopathogens

To confirm whether magnolol and honokiol inhibit pathogenic fungi, their antifungal activity was tested on six fungal pathogens, and the results are shown in Table 3. We found that the magnolol (0.1 mg/mL) was effective against *P. expansum* (Link) Thom, *A. dauci* f.sp. Solani, *F. moniliforme* J. Sheld, *F. oxysporum* Schltdl, *V. mali* Miyabe & G. Yamada, and *R. solani* J.G. Kühn, with 80, 70, 79, 76, 100, and 57% growth inhibition, respectively. The honokiol (0.1 mg/mL) treatment had potent inhibitory effect on the above pathogens with the growth inhibition of 81, 80, 82, 89, 100, and 68%, respectively.

These findings are in good agreement with the antifungal activity of magnolol and honokiol on *Pyricularia oryzae*, *Phytophthora infestans*, *Puccinia recondita*, and *Colletotrichum capsici* [15]. Thus, we conclude that magnolol and honokiol are potential antifungal agents against the plant fungal diseases.

## 3. Materials and Methods

### 3.1. Chemicals and Media

Acetonitrile (CH_3_CN) and pyridine were purchased from Sigma Chemical Co. (San Francisco, CA, USA). Tween-20 was from Coolaber technology Co. LTD (Beijing, China). Glucose and agar were from Shanghai Bioindustrial Engineering Co. Ltd (Shanghai, China) to prepare the PDA (potato dextrose agar) medium. Methanol, petroleum ether, quicklime, and hydrochloric acid were supplied from Sinopharm Group Co. LTD (Beijing, China). Chloroform was from Sulebao Technology Co. LTD (Beijing, China). Eugenol was purchased from Mckuin Biological Co. LTD (Shanghai, China). All chemicals were of analytical grade. Carbendazim was from Zhongxun Agrochemicals Co. LTD (Jiangxi, China). *M. officinalis* bark was purchased from the Jihetang Pharmacy (Yangling, China) and has been identified by Professor Xiaoqian Mu at Northwest A & F University, Yangling, Shanxi, China.

### 3.2. Fungal Strains

*A. alternata* (Fr.) Keissl, *R. solani* J.G. Kühn, *F. moniliforme* J. Sheld, *A. dauci* f.sp. Solani, and *P. expansum* (Link) Thom were obtained from Microbiology Institute of Shaanxi, Shaanxi Province, China. *V. mali* Miyabe & G. Yamada was provided by the State Key Laboratory of Crop Stress Biology for Arid Areas, Northwest A & F University, Yangling, Shanxi, China. *F. oxysporum* Schltdl. was provided by the Laboratory of Phytopathology, Inner Mongolia Agricultural University, Huhhot Inner Mongolia, China. All fungi were grown on PDA medium at 25 ± 2 °C.

### 3.3. Isolation and Purification of Active Compounds and Structure Analysis

The stem bark of *M. officinalis* was rolled up into a column, brown and spicy (Figure 6). With the method of alkali-extraction and acid-precipitation, 100 g of dried bark was ground into powder by a disintegrator XL600B (Xiaobao Electric Co. LTD, Yongkang, China), and extracted three times with 90% methanol. The resulting crude extract (1500 mL) was dried to a paste and then incubated with two times quicklime for 24 h. After washing three times using five times diluted hydrochloric acid, the extract paste was concentrated. After the concentrating step, the extract was dissolved in chloroform and isolated with three volumes of alkaline water (0.5% NaOH solution). The crystals were filtered out from the solution by adjusting the pH value to 7. After drying at a low temperature, the solution that was filtered from the crystal with petroleum for 1 h was naturally cooled to room temperature, and the final product was analyzed by a liquid chromatography-mass spectrometry (LC-MS). The structures were identified by ^1^H and ^13^C-NMR spectra. 1H and ^13^C-NMR spectra were obtained with Bruker Avance III 500 MHz (Bruker Corporation, Karlsruhe, Germany). HPMS were recorded on a Triple TOF 5600+ system (AB SCIEX Co. LTD, Framingham, MA, USA). The high-performance liquid chromatography was performed on an Agilent 1260 liquid chromatography system (Agilent Technologies Co. LTD, Santa Clara, CA, USA) equipped with a Zorbax SB-C18 column, particle size 5 μm, dimension 150 mm × i.d. 9.4 mm, flow rate 7 mL min^−1^.

### 3.4. Antifungal Activity of Magnolol and Honokiol on A. alternata 

Magnolol and honokiol were dissolved in DMSO (1000 mg/mL) and diluted to the required concentration with Tween-20 and distilled water (1:1000 *v*/*v*). Antifungal activities of magnolol and honokiol were tested against *A. alternata* in vitro according to the agar dilution method [34]. Magnolol and honokiol (5 mL) were added to sterilized Petri dishes (90 mm diameter) in concentrations of 0.001, 0.005, 0.01, 0.10, 1.00, 3.00, 5.00, and 7.00 mg/mL, and eugenol (0.1 g/mL) and carbendazim (0.001 g/mL),then were mixed with PDA medium (15 mL). As the control, PDA dishes were added with the same volume of Tween-20 and distilled water (1:1000 *v*/*v*). A 6 mm diameter disc of *A. alternata* mycelia was inoculated into the center of the plate. After incubation for 7 days at 25 ± 2 °C, the colony diameter was measured. Each treatment was performed in triplicate. To calculate the percentage of mycelial growth inhibition (MGI), the following formula was used: mycelium inhibition (%) = [(Dc − Dt)/Dc] × 100, where Dc (cm) is the mean colony diameter for the control Petri dishes, and Dt (cm) is the mean colony diameter for the treatment Petri dishes.

### 3.5. Synergistic Effect Test

Synergistic effect of magnolol and honokiol was tested against *A. alternata.* Magnolol and honokiol (0.1 mg/mL each) were mixed at volume ratios of 1:0, 1:1, 1:4, 1:9, 9:1, 4:1, and 0:1 in a total volume of 5 mL and added to sterilized Petri dishes (90 mm diameter) containing PDA medium (15 mL) and Tween-20 (1:1000 *v*/*v*). A 6 mm diameter disc of *A. alternata* mycelia was inoculated into the center of the plate. After incubation for 7 days at 25 ± 2 °C, the colony diameter was measured. Each treatment was performed in triplicate.

### 3.6. Scanning Electron Microscopy (SEM) Analysis

*A. alternata* incubated on PDA containing magnolol (0.1 mg/mL) and honokiol (0.1 mg/mL) for 7 days were used for SEM analysis. Slices of 7 × 5 × 3 mm segments were cut from the plates and washed three times using normal saline. After keeping the mycelium in 2.5% glutaraldehyde for 2 h at 4 °C, samples were washed with 0.1 M phosphate buffer (pH 7.0) four times. The samples were dehydrated in ethanol sequentially with 30, 50, 70, and 90% ethanol for 20 min each wash, and the final wash was absolute ethyl alcohol for 30 min. Mycelium was observed under a scanning electron microscope after drying. All mycelium samples were viewed with a Nova NanoSEM 450 (FEI Co. LTD, Hillsboro, OR, USA) at 5.00 kV of magnification.

### 3.7. Cell Membrane Permeability Test

The cell membrane permeability test of *A. alternata* was performed, as described previously by Lee et al. [35]. *A. alternata* suspension was incubated in a PDB (potato dextrose broth) medium and rotated with 160 rpm at 25 °C for 48 h. Magnolol (0.1 mg/mL) and honokiol (0.1 mg/mL) were then added to the suspension, 5 mL of which were centrifuged at 10,000 rpm for 15 min. The supernatant conductivity was tested by a conductometer at the start of the measurement (0 h) and then every 1 h, up to 6 h. Each sample was repeated three times.

### 3.8. Antifungal Activity Test

Effects of magnolol and honokiol on the mycelial growth of *A. alternata* (Fr.) Keissl, *P. expansum* (Link) Thom, *F. moniliforme* J. Sheld, *F. oxysporum* Schltdl., *V. mali* Miyabe & G. Yamada, and *R. solani* J.G. Kühn were measured. Different ratios of magnolol and honokiol in a total volume of 5 mL were added to sterilized Petri dishes (90 mm diameter) as indicated in Table 2 and mixed with PDA medium (15 mL). The same volume of distilled water was added to control PDA dishes instead of magnolol and honokiol. A 6 mm diameter disc of pathogens mycelia was inoculated into the center of the plate. After incubation for 7 days at 25 ± 2 °C. The colony diameter was measured as described in Section 3.4 Each treatment was repeated three times.

### 3.9. Statistical Analysis

All data were presented as the mean ± SD by measuring four independent replicates. The data were analyzed using a Data Processing System 15.10 (Hefei, China). The significance of the statistical differences between four means was determined using Duncan′s new complex range method at the 5% level.

## 4. Conclusions

In this study, magnolol and honokiol were purified from *M. officinalis* and identified by ^1^H- and ^13^C-NMR and HP-MS. Magnolol and honokiol showed antifungal activity against *A. alternata* and other phytopathogenic fungi assayed in this study. The antifungal activity can be associated with the hyphal distortion that resulted from the disruption of the cell membrane integrity. Therefore, magnolol and honokiol could be potentially used as antifungal agents against plant fungal pathogens.

## Figures and Tables

**Figure 1 molecules-24-02140-f001:**
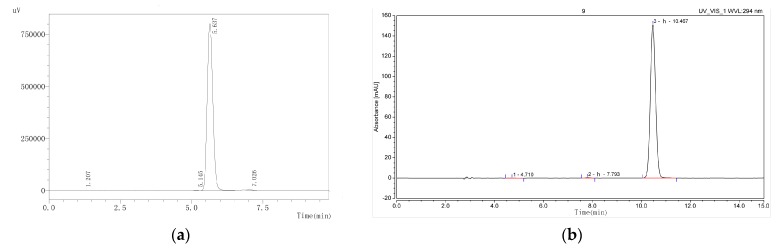
The HPLC chromatogram of magnolol and honokiol. (**a**) Magnolol; (**b**) honokiol.

**Figure 2 molecules-24-02140-f002:**
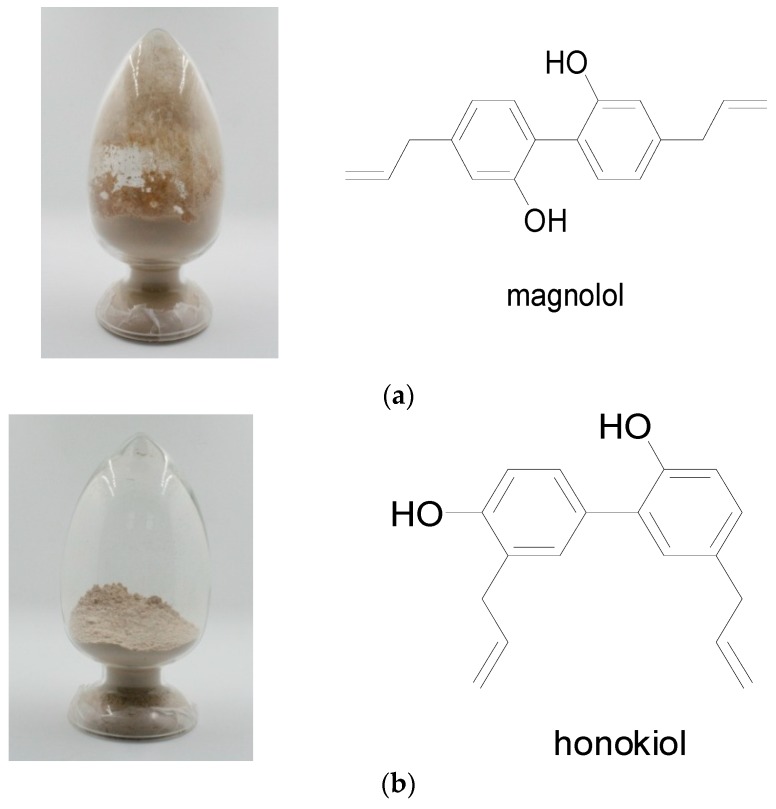
Sample and structure of the compound identified as magnolol and honokiol. (**a**) Magnolol; (**b**) honokiol.

**Figure 3 molecules-24-02140-f003:**
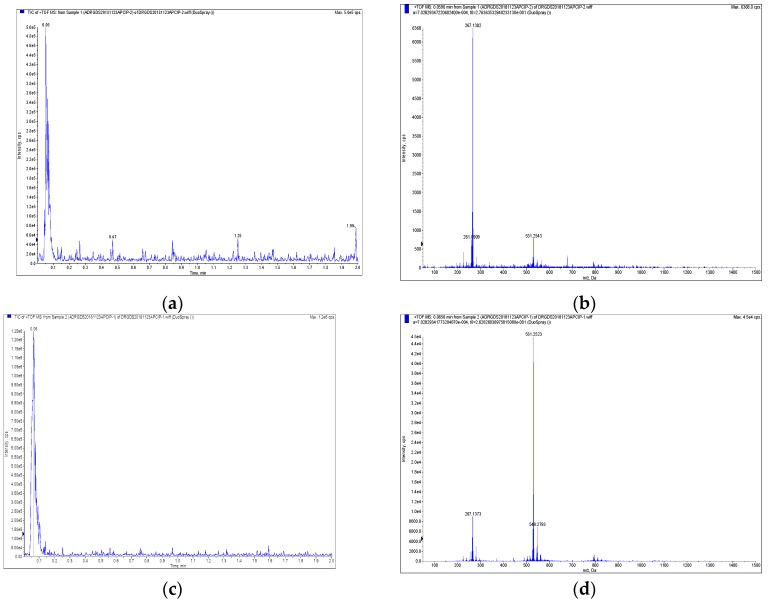
The identification of magnolol and honokiol by high-performance liquid chromatography/mass spectrometry (HPLC/MS) chromatogram. (**a**,**b**) Magnolol; (**c**,**d**) honokiol.

**Figure 4 molecules-24-02140-f004:**
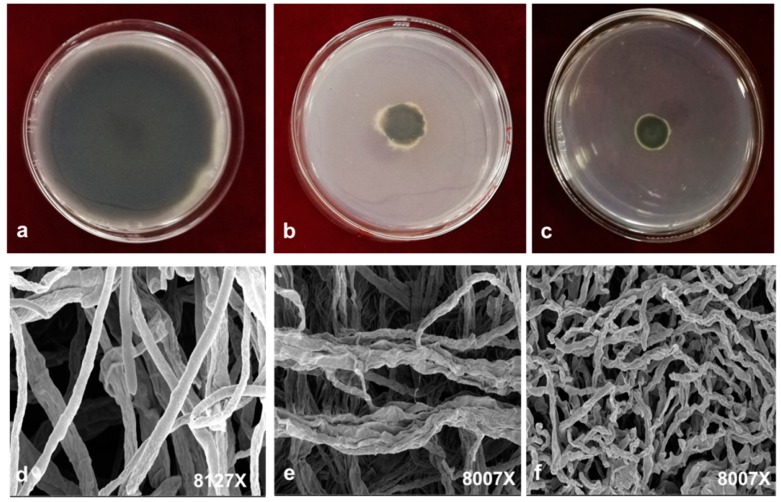
Observation of hyphae morphology of *A. alternata* by electron microphotography. The colony of *A. alternata* on PDA plates is shown in (**a**–**c**): (**a**). Control PDA plate; (**b**). PDA plate containing magnolol (0.1 mg/mL); (**c**). PDA plate containing honokiol (0.1 mg/mL). Hyphae morphology of *A. alternata* on the PDA plate is shown in (**d**–**f**): (**d**). Control PDA plate; (**e**). PDA plate containing the magnolol (0.1 mg/mL); (**f**). PDA plate containing honokiol (0.1mg/mL).

**Figure 5 molecules-24-02140-f005:**
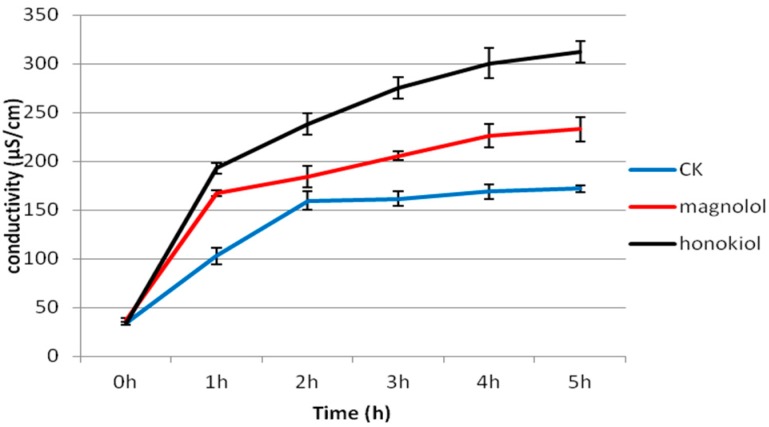
Effects of magnolol and honokiol on the extracellular conductivity of *A. alternata*, CK:PDB medium containing the same volume of Tween-20 and distilled water (1:1000 *v*/*v*).

**Figure 6 molecules-24-02140-f006:**
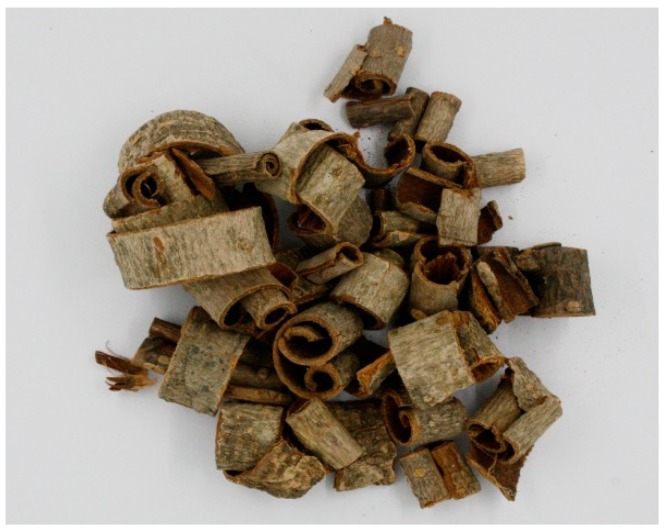
The sample of *M. officinalis* bark.

**Table 1 molecules-24-02140-t001:** Inhibition of magnolol and honokiol at the different concentration on mycelial growth of *A. alternata*.

Concentration (mg/mL)	Mycelial Growth Inhibition (%)			
	Magnolol	Honokiol	Eugenol	Carbendazim
0	0	0	-	-
0.001	7 ± 3.18	9 ± 0.81	-	58 ± 4.64
0.005	16 ± 3.98	20 ± 2.29	-	
0.01	23 ± 2.88	26 ± 1.63	-	-
0.1	77 ± 2.09	91 ± 1.56	11 ± 4.02	-
1	90 ± 3.02	100 *	-	-
3	100 *	100 *	-	-
5	100 *	100 *	-	-
7	100 *	100 *	-	-

Values are presented as mean ± SD. (-) not tested. * significantly different at *P* < 0.05 level by Duncan′s new multiple range test.

**Table 2 molecules-24-02140-t002:** The synergistic effect of magnolol and honokiol on *A. alternata*.

The Volume Ration of Magnolol (0.1 mg/mL) and Honokiol (0.1 mg/mL)	Mycelial Growth Inhibition (%)
1:0	77 ± 2.09
1:1	84 ± 3.85
1:4	89 ± 2.77 *
1:9	88 ± 1.19
9:1	87 ± 1.74
4:1	80 ± 2.11
0:1	91 ± 1.56

Values are presented as mean ± SD. * significantly different at *P* < 0.05 level by Duncan′s new multiple range test.

**Table 3 molecules-24-02140-t003:** The inhibitory efficiency of magnolol and honokiol against six phytopathogens.

Pathogen	Mycelial Growth Inhibition Rate (%)	
	Magnolol	Honokiol
	(0.1 mg/mL)	(0.1 mg/mL)
*P. expansum* (Link) Thom	80 ± 3.95 *	81± 0.23
*Alternaria dauci* f.sp. solani	70 ± 5.26	80 ± 9.23 *
*F. moniliforme* J. Sheld	79 ± 0.69	82 ±0.69 *
*F. oxysporum* Schltdl.	76 ± 1.54 *	89 ± 1.55
*V. mali* Miyabe & G. Yamada	100 *	100 *
*R. solani* J.G. Kühn	57 ± 2.74	68 ± 1.89

Values are presented as mean ± SD. * significantly different at *P* < 0.05 level by Duncan′s new multiple range test.

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
