# Peer review of "Antifungal Effect of Magnolol and Honokiol from Magnolia officinalis on Alternaria alternata Causing Tobacco Brown Spot"

_molecules, 2019, doi:10.3390/molecules24112140_

Round 1

Reviewer 1 Report

Dear Editor

I think that authors had improved their manuscript according the reviewer ´s suggestion. All my comments had been answered, but the manuscript still need some correction before it should be considered for publication in your Journal and the language should be improved and corrected by native speaker.

My comments to authors:

Line 105 I don´t understand the terminus mycelial radial growth inhibition rate – I think the sentence should be changed „The growth inhibition of A.alternata  was 77% in the presence of 0,1mg/ml of magnolol  resp. 91% in the presence of honokiol. 

Line 107 authors had written: And the antifungal activity was much better than 0.1 mg/mL eugenol (11%) and ? 0.001 mg/mL carbendazim (58%)?.I cannot agree with such a claim because carbendazole is more effective than magnolol resp. honokiol. It is shown in Tab.1 that at concentration of 0.001mg /mL carbendazol was able to inhibit the growth of A. alternata on 58% and the obtained inhibition of test compounds was only7% respectively 9%. Could be the effectivity of compounds compared in concentration expressed as mg/mL. „Would it not be better to compare the activity of compounds when their concentrations are expressed in mol / L ?“

Line 115  Chapter 2.3.   So was there any synergic effect of these two compounds or not ? Authors should – improve  English line 116-123

This sentences are realy hard to understand.: line 119-121 The results showed that the  mycelial growth was better than other 6 volume ratio, with a radial growth inhibition of 91%, under  the condation of the volume ratio of magnolol and honokiol(0:1),. When the volume ratio of  magnolol and honokiol was 1:4, 89% of mycelial growth was inhibited. When the volume ratio was  1:0, the growth inhibition was only 77%.

 Table 2. I don ´t understand the term concentration ratio of magnolol(0.1 mg/mL) and honokiol (0.1 mg/mL)  For example the ratio 1:4; what is the concentration of magnolol a what is the concentration of honokiol? Should it not be the volume ratio 1:4?

Line 126  Change the title  2.4. The Inhibition Effects of Magnolol and Honokiol on Morphology and Mycelial of A. alternata.   Inhibition effect on morphology?  I recomend : Growth inhibition and the effect on morphology...

Figure 1 Indication of magnification is missing

Lines 151 -152 Is the conductivity unit correct us/cm u- stays for micro? Should it be not indicated with the Symbol „µ“ and should not the  “s “ not written with capital  „S“ for Siemens

Please write in vitro in italic fond, in whole text

177  change alternate to alternata

187 two comma

Explain  the paragrapgh line 257-258Magnolol and honokiol (5 mL) were added to sterilized Petri dishes (90 mm diameter) to the concentrations of 0.001, 0.005, 0.01, 0.10, 1.00, 3.00, 5.00, 7.00 mg/mL, ?

and 5 mL meugenol  (0.1g/mL) and carbendazim (0.001g/mL) mixed with PDA medium (15 mL). „

Were added to concentration??? And mixed with 15mL PDA???  I am sorry but I realy don ´t understand and have more questions  “What  was the real final concentration of magnolol or honokil in the growth medium? Do I understand it correct?  For example if  5mL of  magnolol (7mg/mL) was dilluted with 15mL of PDA?It was diluted 4 times so the concentration in the growth medium is not 7mg/mL but 4 times lower? Is my consideration correct or not?

Line 257 change meugenol to eugenol

Line 260  and answer for comment 13 of Reviewer 1. Why was diluted Tween20 used for control experiment instead the solvent DMSO? When authors had used 5ml of tested compounds dissolved in DMSO and 15ml of PDA was added, there was high content of DMSO in the growth medium. It is known that DMSO inhibits the growth of fungi (max 2% of DMSO could be used without growth inhibition of fungal cells) moreover DMSO is causing membrane damage.

Line262 After incubation for 7 days at 25 ± 2℃ for 7 days, the colony diameter measured. "was“ is missing in the sentence.

Chapter.4.4; 4.9  Synergism; Antifungal test. I see the same problem with DMSO as  in 4.3

Chaper 4.3 and 4.9 could be merged.

Author Response

Derong An

College of Plant Protection and Shaanxi Key Laboratory of Molecular Biology for Agriculture,

Northwest A&F University,

Yangling, Shanxi, China

Tel: +86-158-2909-7529; Fax: +86-158-2909-7529

E-mail address: [email protected]

                                                                                                         May 27, 2019

Dear Editor,

     We would like to resubmit the revised manuscript entitled “Antifungal effect of magnolol and honokiol from Magnolia officinalis on Alternaria alternata causing tobacco brown spot” to your journal, “Molecules”. We sincerely thank for the reviewers comments, which make a great improvement for our manuscript. We were also very pleased to see that all reviewers recognized the novelty and potential significance of our work. We have added some new data supposed by reviewers, described in detail below, and revised the manuscript to address reviewers’ comments. Here are our point-by-point responses:

Response to Reviewer 1 Comments

Point 1:

I think that authors had improved their manuscript according the reviewer ´s suggestion. All my comments had been answered, but the manuscript still need some correction before it should be considered for publication in your Journal and the language should be improved and corrected by native speaker.

Response:

Considering the Reviewer’s suggestion, the manuscript has been corrected and improved by native speaker.

Point 2:

Line 105 I don´t understand the terminus mycelial radial growth inhibition rate – I think the sentence should be changed „The growth inhibition of A.alternata  was 77% in the presence of 0,1mg/ml of magnolol  resp. 91% in the presence of honokiol.  

Response:

We have revised this part according to the Reviewer’s suggestion.

The growth inhibition of A. alternata was 77% and 91% in the presence of 0.1 mg/ml of magnolol and honokiol, respectively.

Point 3:

Line 107 authors had written: And the antifungal activity was much better than 0.1 mg/mL eugenol (11%) and ? 0.001 mg/mL carbendazim (58%)?.I cannot agree with such a claim because carbendazole is more effective than magnolol resp. honokiol. It is shown in Tab.1 that at concentration of 0.001mg /mL carbendazol was able to inhibit the growth of A. alternata on 58% and the obtained inhibition of test compounds was only7% respectively 9%. “

Response:

Considering the Reviewer’s suggestion, we have corrected as follow:  

The antifungal activity of carbendazim was much higher than that of magnolol and honokiol (Table 1), when used at concentration with 0.001 mg/mL.

Moreover, the antifungal activity was much higher than eugenol (0.1 mg/mL), which showed 11% inhibition of mycelial growth. On the basis of the present results which are also in agreement with the reported by Edris and Farrag (2003), eugenol was completely inactive on Sclerotinia sclerotiorum (Lib), Rhizopus stolonifer (Ehrenb. exFr) Vuill and Mucor sp. (Fisher), but mixing eugenol and linalool in a ratio similar to their concentrations in the original oil was found to enhance the antifungal properties oil indicating a synergistic effect[29].

Point 4:

Line 107 authors had written: Could be the effectivity of compounds compared in concentration expressed as mg/mL. „Would it not be better to compare the activity of compounds when their concentrations are expressed in mol / L ?“

Response:

Thanks for the precious comment and suggestion proposed by the reviewer 1. We understand that it is better to compare the activity of compounds when their concentrations are expressed in mol / L, this may better reveal the possible antifungal activity of magnolol and honokiol. However, according to the specification of carbendazim, it was expressed as mg/mL.

Point 5:

Line 115  Chapter 2.3.   So was there any synergic effect of these two compounds or not ?

Response:

We are very sorry for our poor writing. We have corrected “The results showed that there was not any synergic effect of these two compounds.”

Point 6:

Authors should – improve  English line 116-123

This sentences are realy hard to understand.: line 119-121 The results showed that the  mycelial growth was better than other 6 volume ratio, with a radial growth inhibition of 91%, under  the condation of the volume ratio of magnolol and honokiol(0:1),. When the volume ratio of  magnolol and honokiol was 1:4, 89% of mycelial growth was inhibited. When the volume ratio was  1:0, the growth inhibition was only 77%.

 Table 2. I don ´t understand the term concentration ratio of magnolol(0.1 mg/mL) and honokiol (0.1 mg/mL)  For example the ratio 1:4; what is the concentration of magnolol a what is the concentration of honokiol? Should it not be the volume ratio 1:4?

Response:

Considering the Reviewer’s suggestion, we have corrected as follow:

In order to examine the synergistic effect of a combination of two compounds on the mycelial growth of A. alternata, we combined two compounds in different ratios and tested their effect in vitro assays (Table 2). The results showed that the mycelial growth inhibition was higher than other six volume ratios, with radial growth inhibition of 91%, under the condition of the volume ratio of magnolol and honokiol (0:1). When the volume ratio of magnolol and honokiol was 1:4, a growth inhibition of 89% of mycelial growth was inhibited. When the volume ratio was 1:0, the growth inhibition was only 77%. The results showed that there was not any synergic effect of these two compounds.

Point 7:

Table 2. I don ´t understand the term concentration ratio of magnolol(0.1 mg/mL) and honokiol (0.1 mg/mL)  For example the ratio 1:4; what is the concentration of magnolol a what is the concentration of honokiol? Should it not be the volume ratio 1:4?

Response:

We are very sorry for our poor English.Synergistic effect of magnolol and honokiol was tested against A. alternata. magnolol and honokiol (0.1 mg/mL each) were mixed at volume ratio of 1:0, 1:1, 1:4, 1:9, 9:1, 4:1 and 0:1 in a total volume of 5 mL, and added to sterilized Petri dishes (90 mm diameter) containing PDA medium (15 mL).

Point 8:

Line 126  Change the title  2.4. The Inhibition Effects of Magnolol and Honokiol on Morphologyand Mycelial of A. alternata.   Inhibition effect on morphology?  I recomend : Growth inhibition and the effect on morphology...

Response:

We have made correction according to the Reviewer’s comments.

Point 9:

Figure 1 Indication of magnification is missing

Response:

We are very sorry for our incorrect writing, and indication of magnification has been added.

Point 10:

Lines 151 -152 Is the conductivity unit correct us/cm u- stays for micro? Should it be not indicated with the Symbol „µ“ and should not the  “s “ not written with capital  „S“ for Siemens

Please write in vitro in italic fond, in whole text

Response:

We are very sorry for our incorrect writing, and the conductivity unit has been corrected “µS/cm”, and in whole text it was writed in vitro in italic fond.

Point 11:

177  change alternate to alternate

Response:

We are very sorry for our incorrect writing, and alternate has been corrected alternata.

Point 12:

187 two comma

Response:

We are very sorry for our incorrect writing, and one comma has been deleted.

Point 13:

Explain  the paragrapgh line 257-258 „Magnolol and honokiol (5 mL) were added to sterilized Petri dishes (90 mm diameter) to the concentrations of 0.001, 0.005, 0.01, 0.10, 1.00, 3.00, 5.00, 7.00 mg/mL, ?

and 5 mL meugenol  (0.1g/mL) and carbendazim (0.001g/mL) mixed with PDA medium (15 mL). „

Were added to concentration??? And mixed with 15mL PDA???  I am sorry but I realy don ´t understand and have more questions  “What  was the real final concentration of magnolol or honokil in the growth medium? Do I understand it correct?  For example if  5mL of  magnolol (7mg/mL) was dilluted with 15mL of PDA?It was diluted 4 times so the concentration in the growth medium is not 7mg/mL but 4 times lower? Is my consideration correct or not?

Response:

We are very sorry for our poor English. The reviwer consideration is correct. For example if 5mL of magnolol (7mg/mL) was dilluted with 15mL of PDA, It was diluted 4 times so the concentration in the growth medium is not 7mg/mL but 4 times lower. We has been corrected as follow:

Magnolol and honokiol (5 mL) were added to sterilized Petri dishes (90 mm diameter) to the concentrations of 0.001 mg/mL, 0.005 mg/mL, 0.01 mg/mL, 0.10 mg/mL, 1.00 mg/mL, 3.00 mg/mL, 5.00 mg/mL, 7.00 mg/mL, and 5 mL eugenol (0.1 g/mL) and carbendazim (0.001 g/mL) mixed with PDA medium (15 mL).

Point 14:

“Line 257 change meugenol to eugenol”

Response:

We are very sorry for our incorrect writing, and meugenol has been corrected eugenol.

Point 15:

“Line 260  and answer for comment 13 of Reviewer 1. Why was diluted Tween20 used for control experiment instead the solvent DMSO? When authors had used 5ml of tested compounds dissolved in DMSO and 15ml of PDA was added, there was high content of DMSO in the growth medium. It is known that DMSO inhibits the growth of fungi (max 2% of DMSO could be used without growth inhibition of fungal cells) moreover DMSO is causing membrane damage.”

Response:

We are very sorry for our incorrect writing. We searched scientific literature and knowed moreover DMSO is causing membrane damage. Magnolol and honokiol were dissolved in DMSO (1000 mg/mL), and diluted to the required concentration with Tween-20 and distilled water (1:1000 v/v).Forexample, 5 mL honokiol (5 mL/mL) were added to sterilized Petri dishes (90 mm diameter) mixed with PDA medium (15 mL). In the 20 ml solution, DMSO content just was 0.1%, it can be ignored. So diluted Tween20 was used for control experiment instead the solvent DMSO.

Point 16:

“Line262 After incubation for 7 days at 25 ± 2 for 7 days, the colony diameter measured. "was is missing in the sentence.”

Response:

We are very sorry for our incorrect writing, and was has been added.

Point 17:

“Chapter.4.4; 4.9  Synergism; Antifungal test. I see the same problem with DMSO as  in 4.3”

Response:

We are very sorry for our incorrect writing. We searched scientific literature and knowed moreover DMSO is causing membrane damage. Magnolol and honokiol were dissolved in DMSO (1000 mg/mL), and diluted to the required concentration with Tween-20 and distilled water (1:1000 v/v).Forexample, 5 mL honokiol (5 mL/mL) were added to sterilized Petri dishes (90 mm diameter) mixed with PDA medium (15 mL). In the 20 ml solution, DMSO content just was 0.1%, it can be ignored. So diluted Tween20 was used for control experiment instead the solvent DMSO.

Point 18:

“Chaper 4.3 and 4.9 could be merged.”

Response:

According to the Reviewer’s suggestion, the alkaline water in the revised draft is as follow:

3.3. Isolation and purification of active compounds and structure analysis

The stem bark of M. officinalis, was rolled up into a column, brown and spicy (Figure 6). With the method of alkali-extraction and acid-precipitation, 100 g dried bark was ground into powder by disintegrator XL600B (Xiaobao Corporation, China), and extracted three times with 90% Methanol. The resulted crude extract (1500 mL) was dried to a paste and then incubated with two times quicklime for 24 h. After washing three times using five times diluted hydrochloric acid, the extract paste was concentrated. After the concentrating step, the extract was dissolved in chloroform and isolated with three volumes of alkaline water (0.5% NaOH solution). The crystals were filtered out from the solution by adjusting for pH value to 7. After drying at a low temperature, the solution that was filtered from the crystal with petroleum for 1h was naturally cooled to room temperature, and the final product was analyzed by a liquid chromatography-mass spectrometry (LC-MS). The structures were identified by 1H and 13C NMR spectra. 1H and 13C NMR spectra were obtained with Bruker Avance III 500 MHz (Bruker Corporation, Karlsruhe, Germany). HPMS were recorded on a Triple TOF 5600+ system (AB SCIEX, USA). The high-performance liquid chromatography was performed on an Agilent 1260 liquid chromatography system equipped with a Zorbax SB-C18 column (particle size 5 μm, dimension 150 mm × i.d. 9.4 mm, flow rate 7 mL min-1, Agilent Technologies) .

Thank you for your consideration of our manuscript.

Yours sincerely,

Yahan Chen, Ph.D.

Reviewer 2 Report

The followings are my comments and suggestions for the authors:

A minor consideration is the English language which needs certain re-write.

However, the manuscript exhibits several weaknesses identified below and also in the attached file with more details (it is not a complete list).

In section 2.1.Chemistry:

1.     Both structures of magnolol and honokiol should be enlarged and do not be deformed.

In section 2.2. Antifungal Activity of Magnolol and Honokiol on A. alternata:

1.     These two antifungal control reference compounds should be given a brief introduction in the text.

2.     Lines 105-107: This statement should be corrected.
When the concentration was 0.001 mg/mL, the antifungal activity of carbendazim was much better than magnolol and honokiol (Table 1).

3.     In Table 1.: The title should be corrected as “Inhibition of magnolol and honokiol at different concentration on mycelial growth of A. alternata”.

In section Discussion:

There are many sentences in the discussion that are similar to the results, and some sentences can be placed in the introduction or conclusion. Therefore, it is highly recommended the manuscript sections of the article are presented as the followings:

1. Introduction, 2. Results and Discussion, 3. Materials and Methods, 4. Conclusions

In section References:

References 2, 4, 5, 11, 12, 15 and 25 have some errors.

Yours

Author Response

Derong An

College of Plant Protection and Shaanxi Key Laboratory of Molecular Biology for Agriculture,

Northwest A&F University,

Yangling, Shanxi, China

Tel: +86-158-2909-7529; Fax: +86-158-2909-7529

E-mail address: [email protected]

                                                                                                         May 27, 2019

Dear Editor,

     We would like to resubmit the revised manuscript entitled “Antifungal effect of magnolol and honokiol from Magnolia officinalis on Alternaria alternata causing tobacco brown spot” to your journal, “Molecules”. We sincerely thank for the reviewers Points, which make a great improvement for our manuscript. We were also very pleased to see that all reviewers recognized the novelty and potential significance of our work. We have added some new data supposed by reviewers, described in detail below, and revised the manuscript to address reviewers’ Points. Here are our point-by-point responses:

Response to Reviewer 2 Points

Point 1:

“T A minor consideration is the English language which needs certain re-write.

Response

Considering the Reviewer’s suggestion, the manuscript has been corrected and improved by native speaker.

Point 2:

“In section 2.1.Chemistry:

1.     Both structures of magnolol and honokiol should be enlarged and do not be deformed.”

Response

Considering the Reviewer’s suggestion, we have enlarged the structures.

 (a)

(b)

Figure 2. Sample and structure of the compound identified as magnolol and honokiol, (a) magnolol; (b) honokiol.

Point 3:

“In section 2.2. Antifungal Activity of Magnolol and Honokiol on A. alternata:

1.     These two antifungal control reference compounds should be given a brief introduction in the text.”

Response

We have revised this part according to the Reviewer’s suggestion as follow:

Carbendazim is a systemic fungicide with both protective and curative activities against a wide range of fungal diseases[27]. Its antifungal activity was much higher than that of magnolol and honokiol (Table 1), when used at concentration with 0.001 mg/mL. However, it has been reported that carbendazim, a endocrine disrupters, has mutagenic and teratogenic effects on mammals at low doses[28].

The mycelial radial growth inhibition rates were 77% and 91% for magnolol and honokiol, respectively, when the concentration of each was 0.1 mg/mL. Moreover, the antifungal activity was much higher than eugenol (0.1 mg/mL), which showed 11% inhibition of mycelial growth. On the basis of the present results which are also in agreement with the reported by Edris and Farrag (2003), eugenol was completely inactive on Sclerotinia sclerotiorum (Lib), Rhizopus stolonifer (Ehrenb. exFr) Vuill and Mucor sp. (Fisher), but mixing eugenol and linalool in a ratio similar to their concentrations in the original oil was found to enhance the antifungal properties oil indicating a synergistic effect[29].

Point 4:

“.     Lines 105-107: This statement should be corrected.

When the concentration was 0.001 mg/mL, the antifungal activity of carbendazim was much better than magnolol and honokiol (Table 1).”

Response

Considering the Reviewer’s suggestion, we have corrected the statement.

When the concentration was 0.001 mg/mL, the antifungal activity of carbendazim was much better than magnolol and honokiol (Table 1).”

Point 5:

“In Table 1.: The title should be corrected as “Inhibition of magnolol and honokiol at different concentration on mycelial growth of A. alternata””

Response

Considering the Reviewer’s suggestion, we have corrected.

Table 1. Inhibition of magnolol and honokiol at the different concentration on mycelial growth of A. alternata.

Concentrationmg/mL

Mycelial   growth inhibition%

Magnolol

Honokiol

Eugenol

Carbendazim

0

0

0

-

-

0.001

7±3.18

9±0.81

-

58±4.64

0.005

16±3.98

20±2.29

-

0.01

23±2.88

26±1.63

-

-

0.1

77±2.09

91±1.56

11±4.02

-

1

90±3.02

100*

-

-

3

100*

100*

-

-

5

100*

100*

-

-

7

100*

100*

- 

- 

Values are presented as mean ± SD.

(-) indicates not tested.

* indicates significantly different at P0.05 level by Duncan’s new multiple range test.

Point 6:

“In section Discussion:

There are many sentences in the discussion that are similar to the results, and some sentences can be placed in the introduction or conclusion. Therefore, it is highly recommended the manuscript sections of the article are presented as the followings:

1. Introduction, 2. Results and Discussion, 3. Materials and Methods, 4. Conclusions”

Response

We have made correction according to the Reviewer’s Points in the manuscript. The manuscript sections of the article are presented as the followings:

1. Introduction, 2. Results and Discussion, 3. Materials and Methods, 4. Conclusions”

Point 7:

“In section References:

References 2, 4, 5, 11, 12, 15 and 25 have some errors.”

Response

We have made correction according to the Reviewer’s Points in the manuscript.

References

1.          Lei, Z.Q. The wet medicine. The Chinese Materia Medica, 1st ed.; Shanghai Science and Technology Press: Shanghai, China, 1995; 128.

2.          Choi, J.H.; Ha, J.; Park, J.H.; Lee, J.Y.; Lee, Y.S.; Park, H.J. Costunolide triggers apoptosis in human leukemia U937 cells by depleting intracellular thiols. Cancer Science. 2002, 93, 1327−1333.

3.          Kang, J.S.; Lee, K.H.; Han, M.H.; Lee, H.; Ahn, J.M.; Han, S.B.; et al. Antiinflammatory activity of methanol extract isolated from stem bark of Magnolia kobus. Phytother Res. 2008, 22, 883−888.

4.          Kong, C.W.; Tsai, K.; Chin, J.H.; Chan, W.L.; Hong, C.Y. Magnolol attenuates peroxidative damage and improves survival of rats with sepsis. Shock. 2000, 13, 24−28.

5.          Lee, J.W.; Lee, Y.K.; Lee, B.J.; Nam, S.Y.; Lee, S.I.; Kim, Y.H.; et al. Inhibitory effect of ethanol extract of Magnolia officinalis and 4-O-methylhonokiol on memory impairment and neuronal toxicity induced by beta-amyloid. Pharmacol Biochem Be 2009, 95, 31−40.

6.          Hattori, M.; Bae, K.H.; Tsunezuka, M.; Namba, T. Studies on Dental Caries Prevention by Traditional Chinese Medicines (Part I): Screening of Crude Drugs for Antibacterial Action against Streptococcus mutans. Japanese Journal of Pharmacognosy 1981, 35, 295-302.

7.          Tan Z.X.; Yang L.X. Biological control of tobacco brown spot disease: present and future . Acta Tabacaria Sinica 2005, 11, 34-38.

8.          Tachikawa, E.; Takahashi, M.; Kashimoto, T. Effects of extract and ingredients isolated from Magnolia obovata thunberg on catecholamine secretion from bovine adrenal chromaffin cells. Biochem Pharmacol. 2000, 60, 433−440.

9.          LaMondia J.A. Outbreak of Brown Spot of Tobacco Caused by Alternaria alternata in Connecticut and Massachusetts. Plant Dis. 2001, 85, 230-230.

10.       Chen, M.C.; Lee, C.F.; Huang, W.H.; Chou, T.C. Magnolol suppresses hypoxia-induced angiogenesis via inhibition of HIF-1alpha/VEGF signaling pathway in human bladder cancer cells. Biochem Pharmacol. 2013, 85, 1278–1287.

11.       Chuang, T.C.; Hu, S.C.; Cheng, Y.T.; Shao, W.S.; Wu, K.; Fang, G.S.; Ou, C.C.; Wang, V. Magnolol down-regulates HER2 gene expression, leading to inhibition of HER2-mediated metastatic potential in ovarian cancer cells. Cancer Lett. 2011, 311, 11–19.

12.       Sun, L.M.; Liao, K.; Liang, S.; et al. Synergistic activity of magnolol with azoles and its possible antifungal mechanism against\r, Candida albicans. J Appl Microbiol. 2015, 118, 826-838.

13.       Lee, Y.J.; Lee, Y.M.; Lee. C.K.; et al. Therapeutic applications of compounds in the Magnolia family. Pharmacology & Therapeutics 2011, 130, 157-176.

14.       Bang, K.H.; Kim, Y.K.; Min, B.S.; Na, M.K.; Rhee, Y.H.; Lee, J.P.; Bae, K.H. Antifungal activity of magnolol and honokiol. Arch Pharm Res. 2000, 23, 46–49.

15.       Choi, N.H.; Choi, G.J.; Min, B.S.; et al. Effects of neolignans from the stem bark of Magnolia obovata on plant pathogenic fungi. J Appl Microbiol. 2010, 106, 2057-2063.

16.       Liu X.M.; Li D.Z. Researching advance of tobacco brown spot caused by Alternaria alternata. Journal of Northeast Agricultural university 2000, 1, 80-85.

17.       Yi L.; Xiao C.G. Advances in studies on control of tobacco brown spot. plant protection 2003, 5, 10-14.

18.       Li, S.Y.; Guan, B.Q.; Wei, F.J. The major disease of tobacco. Technical Guide for Tobacco Production. 1th ed.; China Agriculture Press: Beijing, China, 2010; 169-170.

19.       Chen, C.R.; Tan, R.; Qu, W.M.; Wu, Z.; Wang, Y.; Urade, Y.; Huang, Z.L. Magnolol, a major bioactive constituent of the bark of Magnolia officinalis, exerts antiepileptic effects via the GABA/benzodiazepine receptor complex in mice. Brit J Pharmacol. 2011, 164, 1534–1546.

20.       Chen, J.; Li, L.Y.; Gao, M.; Ma, G.H.; Chen, G.K.; Yu, Q.T. Indoor Screening and Field Trial of Fungicides against Tobacco Brown Spot. Chinese Tobacco Science 2017, 38, 73-77.

21.       Chang, L.J.; Jian, Z.B.; Xiao, B.H.; Huang, R.H.; Cai, D.S.; Zhang, C.S. Essential oil of Syringa oblata Lindl as a potential biocontrol agent against tobacco brown spot caused by Alternaria alternata. Crop Prot. 2018, 104, 41-46.

22.       Mi, Y.Y.; Byeong, J.C.; Jin, C.K. Recent Trends in Studies on Botanical Fungicides in Agriculture. Plant Pathology J. 2013, 29, 1-9.

23.       Shin, K.S.; Lee, S.; Cha, B. Antifungal activity of plumbagin purified from leaves of Nepenthes ventricosa x maxima against phytopathogenic fungi. Plant Pathol. 2007, 23, 113−115.

24.       Kijjoa, A.; Pinto, M.M.M.; Tantisewie, B.; Herz, W. A biphenyl type neolignan and biphenyl ether from Magnolia henryi. Phytochemistry 1989, 28, 1284–1286.

25.       Ho, K.Y.; Tsai, C.C.; Chen, C.P.; et al. Antimicrobial activity of honokiol and magnolol isolated from Magnolia officinalis. Phytotherapy Research Ptr. 2001, 15, 139-141.

26.       Wang, X.; Wang, Y.Q.; Geng, Y.L.; Li, F.W.; Zheng, C.C. Isolation and purification of honokiol and magnolol from cortex Magnoliae officinalis by high-speed counter-current chromatography. Joural of Chromatography A. 2004, 1036, 171-175.

27.       Boudina, A.; Emmelin, C.; Baaliouamer, A.; Grenier-Loustalot, M.F.; Chovelon, J.M. Photochemical behaviour of carbendazim in aqueous solution. Chemosphere. 2003, 50, 649-655.

28.       Holtman, M.A.; Kobayashi, D.Y. Identification of Rhodococcus erythropolis isolates capable of degrading the fungicide carbendazim. Appl Microbiol Biot. 1997, 47, 578–582.

29.       Edris, A.E. and Farrag, E.S. Antifungal activity of peppermint and sweet basil essential oils and their major aroma constituents on some plant pathogenic fungi from the vapor phase. Nahrung. 2003, 47, 117-121.

30.       Tyagi, A.K.; Malik, A. Antimicrobial potential and chemical composition of Mentha piperita oil in liquid and vapour phase against food spoiling microorganisms. Food Control. 2011, 22, 1707-1714.

31.       Zhou, H.; Tao, N.; Jia, L. Antifungal activity of citral, octanal and α-terpineol against Geotrichum citri-aurantii. Food Control. 2014, 37, 277-283.

32.       Jing, C.L.; Zhao, J.; Han, X.B.; Huang, R.H.; Cai, D.S.; Zhang, C.S. Essential oil of Syringa oblata Lindl. as a potential biocontrol agent against tobacco brown spot caused by Alternaria alternata. Crop Prot. 2018, 104, 41-46.

33.       Bajpai, V.K.; Sharma, A.; Baek, K.H. Antibacterial mode of action of Cudrania tricuspidata fruit essential oil, affecting membrane permeability and surface characteristics of food-borne pathogens. Food Control. 2013, 32, 582-590.

34.       Paul, S.; Dubey, R.C.; Maheswari, D.K.; Kang, S.C. Trachyspermum ammi (L.) fruit essential oil influencing on membrane permeability and surface characteristics in inhibiting food-borne pathogens. Food Control. 2011, 22, 725-731.

35.       Shao, X.; Cheng, S.; Wang, H.; Yu, D.; Mungai, C. The possible mechanism of antifungal action of tea tree oil on Botrytis cinerea. J Appl Microbiol. 2013, 114, 1642-1649.

36.       Turgis, M.; Han, J.; Caillet, S.; Lacroix, M. Antimicrobial activity of mustard essential oil against Escherichia coli O157: H7 and Salmonella typhi. Food Control. 2009, 20, 1073-1079.

37.       Yahyazadeh, M.; Omidbaigi, R.; Zare, R.; Taheri, H. Effect of some essential oils on mycelial growth of Penicillium digitatum Sacc. World J Microb Biot. 2008, 24, 1445-1450.

38.       Abbaszadeh, S.; Sharifzadeh, A.; Shokri, H.; Khosravi, A.R.; Abbaszadeh, A. Antifungal efficacy of thymol, carvacrol, eugenol and menthol as alternative agents to control the growth of food-relevant fungi. J mycol med. 2014, 24, 51–56.

39.       Lee, H.J.; Choi, G.J.; Cho, K.Y. Correlation of lipid peroxidation in Botrytis cinerea caused by dicarboximide fungicides with their fungicidal activity. J Agr Food Chem. 1998, 46, 737-741.

Sincerely thanks for your nice Points.

Yours sincerely,

Yahan Chen, Ph.D.

Round 2

Reviewer 1 Report

The manuscript in improoved and is writen very clear. All my coments are incorporated in the text. I suggest only minor improovment of the manuscript that I have detcted during its reading.

My suggestions:

line 57 against A. alternata. Plumbagin (29).  Something is wrong there

line 67 in vitro – italic

line 116 tab.1  A.alternata  - italic

line 121 in vitro  - italic

Table 2. Is the 1st column heading correct? Concentration ration? Should it not be the volume ration?

line 254 Heading 3.4. the word „Test“ could be deleted

line257 in vitro – italic

line269 Magnolol big M – new sentence

Line 296 Was the control plate not prepared as is  described in the in Chap. 3.4. : „As the control, PDA dishes were added with the same volume of Tween-20 and distilled water (1:1000 v/v)“ .  Is Tween missing in the control plate?

Line 299 „as described in 4.5“? not 3.4? Chap. 4.5 does not exsint

Line 306, 307 – Conclusion : „ Magnolol and honokiol showed antifungal activity against A. alternata.“ I would add at the end of the sentence : and other phytopathogenic fungi assayed in this study.

After incorporating my comments, I recommend to accept the manuscript for publication

This manuscript is a resubmission of an earlier submission. The following is a list of the peer review reports and author responses from that submission.

Round 1

Reviewer 1 Report

The status of available antifungal agents both in therapy as well as in agriculture is very difficult and very hard. As authors had mentioned in the text it is very important to search for new antifungals. The results that the authors have shown are interesting but in my opinion they have forced the antifungal activity of the two compounds. Sometimes less is more. The concentrations that were assign as the values of MIC resp. MFC are too high and in my opinion would be inhibitory  resp. toxic to any cells. In the introduction authors have mentioned lot of bio-effects of metabolites of Magnolia officialis. Is the effect of the assayed compounds really selective even in such high concentrations?  Why have authors shown the MIC and MFC values on example of A. alternata and not on other fungi that are mentioned in the text, as well as the indicated mechanism of action why is it shown only on the example of one model fungus?  Why did not authors assayed all of model fungi mentioned in the text? (The assay is not so money and time consuming that they would be not able to do it).

Moreover the inhibitory effect (MIC, MFC) shown in the Fig. 3 is not well interpreted.  In the Fig.3 there is a colony formed, the inhibition compared to the control is clear but it is not a 100% inhibition. When some compound has 100% of inhibition activity none of the conidia are able to germinate. I would recommend the authors to review their results, evaluate the growth inhibition up to 0.1mg/ml; 77% inhibition of fugal colonies is enough it is very near the MIC80 value that is for the antifungal activity of natural product a good result. Some compounds are not able to inhibit the microbes in higher level even in higher concentration. The concentrations higher than 0.1 mg/ml are in my opinion too high for showing the selective toxicity,  if there are no  toxicity tests done on other eukaryotic cells (for example mammalian cells cultures).

Alternaria alternate -is not correct written - correct the name of filamentous fungus to Alternaria alternata in the whole text

Tab 1., Tab.2. Indexes after the value of growth inhibition – what does it mean - it is not clear enough, if it means a significant effect P>0.05 (as it is written under the Tab.) why  is it  not described as usually by  a star (*) symbol ?

Tab.1 Concentration of tested compounds is missing;

Tab.1, Tab.2 Showing the inhibition % on 2 decimal place is not necessary, the whole No. for example 90%  instead 90,09%

Tab.2 correct the column name concentration –„ i „ missing

Fig.3 the magnification to micro photos is missing – are the photos done with the same magnification d and e seems that the magnification are the same but f , I don ´t think so.

In the micro – photos (SEM) there is clearly seem that the morphology of the hyphae are different under the pressure of tested compounds, it looks like hyphae curling effect (morphogenic effect of some antifungals). I think that there is something more not only the altered permeability of the cell membrane or its damage. If there is an altered permeability of the membrane it should be more sensitive to some hyperosmotic environment. I would recommend the authors to assay it. Also some fluoresce-dyes assays are available to show the effect on membrane damage, or the assay with Beta vulgaris - it is clearly seen the leakage of red color in the presence of tested compounds when is causing membrane damage; resp. some assayed with red blood cells where hemoglobin leakage is  obvious when the membrane is damaged. These assays are not difficult and would nicely support the results with conductivity. Regarding conductivity, is the conductivity unit correct?

Authors had evaluated the growth inhibition after 3 days is the fungus in stationary phase after such short period. We are usually cultivating A. alternata for 7 days until it reaches the stationary phase.

What is the solvent for tested compounds? Was it used in control samples when the antifungal activity was evaluated? It is not clear enough explained in the text.

Did the authors think about the synergy of these two compounds I think interesting results would be obtained when both compounds would be put together and the antifungal assay would be repeated?

The formula for growth inhibition evaluation – is not clear I don ´t understand the symbols for Dt, Dc

Row 215 how was alkaline water prepared?   Row 2015 Correct the PH - for pH

Reviewer 2 Report

This is an interesting area of chemical/microbiological research to search for alternative antimicrobial agents from natural products. The subject falls within the scope of the Journal and the scientific content could be interesting, however the paper is not acceptable for publication in this form. To use natural compounds as possible antimicrobial agents in plant phytotherapy the toxicity tests of the compounds are missing and in my opinion some tests using the whole bark extract are necessary.

Line 3, line 18-19, line 30 Title “Mangnolia” is wrong: please change to Magnolia officinalis (in italic)

Line 3 Alternaria alternata in italic

Organism names should be wrote in full the first time they are cited, then they must abbreviated: i.e. Fusarium moniliforme, then F.oxysporum (in italic).

All organism names should be written in italic

Check Alternaria alternata not Alternaria alternate!!!!!! Check and correct all the names of the fungi

Table 1 and test: the list of fungi is not homogeneous. Fungi from the same family should be placed close together, a logical order should be used:

Penicillium expansum (Link) Thom

Alternaria alternata

Fusarium

Valsa mali

Rhizoctonia

It would also be more appropriate to include the family or the class or phylum. I.e Rhizoctonia is a Basidiomycete

Line 116 How do the authors see that the plates contain Magnolol and Honokiol? I think this is not visually possible. Please modify the sentence or better specify

Authors claim that: “The mycelia of A. alternate were showing distorted, aggregated and partly squashed when growing on the plant containing the magnolol at the MIC of 3 mg/mL (Figure. 3e)”. Figure 3 refers to in vitro tests not in vivo! Please explain better

Lines 149-150 I do not really understand what the authors want to say. Please specify better

Lines 179-180 These results are consistent with the previous study. I mean? please specify better

In the discussion the results obtained should be better analyzed and compared with similar results. Human pathogens and plant pathogens are not comparable if they are completely different species (i.e Trichophyton or Epidermophyton). The authors should better explain that their results are not easily comparable with other studies because the experiments were done using other fungi, or other methods. However, the authors' results indicate e.g. that these molecules have good (or poor) antifungal
activity against plant pathogens as demonstrated for human pathogens .... and
here refer to examples on dermatophytes or other

In the paper test Replace mycelia growth with mycelial growth

Line 268 replace hypha with hyphal

Reviewer 3 Report

My comments and suggestions for the authors are as the followings:

In the title and/or article:

1.     The “Magnolia officinalis” and “Alternaria alternata” should be in italic font.

2.     The “Mangnolia officinalis (M. officinalis)” should be corrected as “Magnolia officinalis (M. officinalis)”.

3.     The “Alternaria alternate (A. alternate)” should be corrected as “Alternaria alternata (A. alternata)”.

4.     The “hypha” should be corrected as “hyphae”.

Abstract:

The names of following six fungal strains are based on MycoBank and Index Fungorum data base.
Alternaria alternata (Fr.) Keissl.
Rhizoctonia solani J.G. Kühn
Fusarium moniliforme J. Sheld.
Penicillium expansum (Link) Thom.
Valsa mali Miyabe & G. Yamada
Fusarium oxysporum Schltdl.

Results:

1.     In section 2.1.: It should be added a brief description about the color, odor, taste of the Magnolia officinalis cortex (Magnolia bark) along with a picture in Figure.

2.     The quantitative and qualitative data from LC-MS is more interesting than purification by HPLC. Therefore, supplying the charts and data of LC-MS in Figure and article is necessary.

3.     If allowed, it’s better to provide elemental analysis data of compounds magnolol and honokiol. The EA data of compounds may support the credibility of the bioactivity analysis values.

4.     In section 2.1. (Lines 75 and 80): C18H19O2 should be corrected as C18H18O2.

5.     Paragraphs 2.2. and 2.3. seem to be opposite; (it may be better to exchange 2.2. to 2.3.).

6.     In section 2.2.: Why don’t you use a natural product, for example eugenol, as an antifungal control reference in this article?

7.     The test concentration of magnolol and honokiol should be noted in the Table 1..

8.     In section 2.3. (Line 113): A clearer description would be: The different letters (a, b, c, d, e, f) in the same column indicate significant ----.

9.     In section 2.4. (Line 118): PDA (potato dextrose agar) plate

10.  In section 2.5.: It should be mentioned the temperature of the cell membrane permeability test.

11.  In section 2.5. (Line 140): Is the unit of conductivity value correct?

12.  The charts in Figure 4. and Lines 144-145: It should be modified the colors of (-):control; (-): MIC; (-): MFC.

Discussion:

Lines 147-160: There sentences are better to be incorporated into the introduction section.

Materials and Methods:

1.     In section 4.1.: It should be mentioned which specialist had identified the Magnolia officinalis bark.

2.     In section 4.3. (Lines 203-207): It should be mentioned the purification method and instrument (state the manufacturer, city and country).

3.     Because this section is “Methods”, it may be better to have the following section title:

4.4. Antifungal Activity Test

4.5. Minimum Inhibitory Concentration (MIC) and Minimal Fungicidal Concentration (MFC) Test

4.6. Scanning Electron Microscopy (SEM) Assay

4.7. Cell Membrane Permeability Test

4.8. Statistical Analysis

4.     Line 157: PDB (potato dextrose broth)

References:

The specifications of the references should comply with the “Instructions to Authors”.

For example: Reference 2.

Choi, J. H.; Ha, J.; Park, J. H.; Lee, J. Y.; Lee, Y. S.; Park, H. J. Costunolide triggers apoptosis in human leukemia U937 cells by depleting intracellular thiols. Japanese Journal of Cancer Research. 2002, (93): 1327−1333.  should be modified as

Choi, J.H.; Ha, J.; Park, J.H.; Lee, J.Y.; Lee, Y.S.; Park, H.J. Costunolide triggers apoptosis in human leukemia U937 cells by depleting intracellular thiols. Jpn J Cancer Res. 2002, 93, 1327-1333.

Yours
